# ROBUSTNESS EVALUATION
# USING LOCAL SUBSTITUTE NETWORKS

## ABSTRACT

The robustness of a neural network against adversarial examples is important when a deep classifier is applied in safety-critical use cases like health care or autonomous driving. To assess the robustness, practitioners use various tools ranging from adversarial attacks to the exact computation of the distance to the decision boundary. We use the fact that the robustness of a neural network is a local property and empirically show that computing the same metrics for the smaller local substitute networks yields reasonable estimates of the robustness for a lower cost. To construct the substitute network, we develop several pruning techniques that preserve the local properties of the initial network around a given anchor point. Our experiments on multiple datasets prove that this approach saves a significant amount of computation and is especially beneficial for larger models.

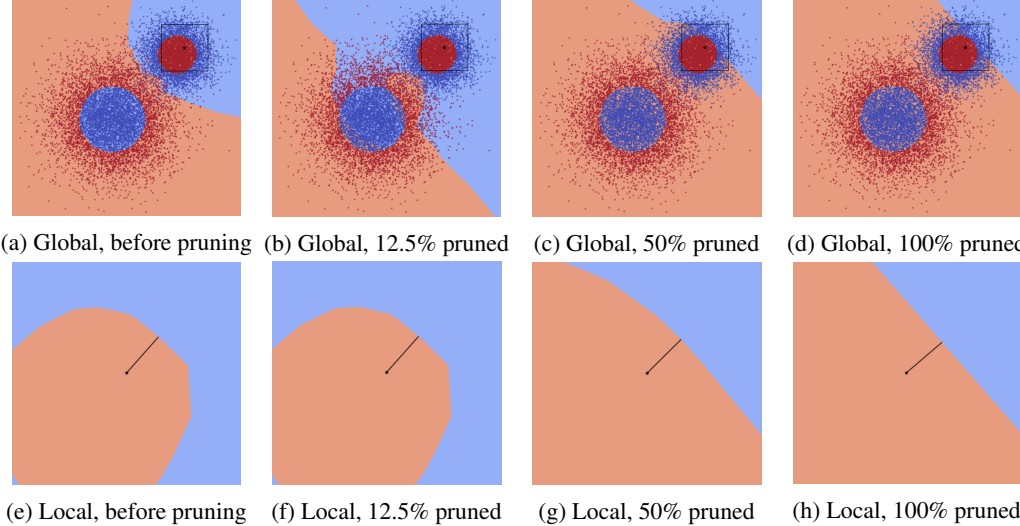

| (a) Global, before pruning | (b) Global, 12.5% pruned | (c) Global, 50% pruned | (d) Global, 100% pruned |

| (e) Local, before pruning | (f) Local, 12.5% pruned | (g) Local, 50% pruned | (h) Local, 100% pruned |

Figure 1: A toy example in the two-dimensional setting. We show the boundaries of a classifier, before (1a and 1e) and after the pruning (1b - 1d and 1f - 1h), when up to the 100% of the hidden neurons are removed. The sample, we are interested in, is marked by the black point and the square around it on the plots with the global view (1a - 1d) shows the local region that is depicted on the other four plots (1e - 1h). While the global behaviour changes a lot, the boundary around the chosen anchor point remains similar. Most importantly, the distance to the closest adversarial from the anchor point (shown by the black line on the plots 1e - 1h) does not change significantly.

## 1 INTRODUCTION

The impressive success of neural networks in various complicated tasks makes them irreplaceable for practitioners despite the known flaws. One problem that continuously gains more attention is the robustness of the deep neural classifiers. While multiple notions of robustness exist, depending on the use case, we consider the most basic concept of robustness against adversarial examples -

small perturbations of the correctly classified samples that lead to a false prediction. The presence of adversarial examples severely limits the application of neural networks in safety-critical tasks like health care and autonomous driving, where the data is collected from sensors, and it is not acceptable that the same image, for example, a road sign, is classified differently depending on the signal noise.

While this problem is widely known, formal robustness verification methods do not allow for assessing the classifier's robustness when the network is large or require specific modifications in the network's architecture or training procedure. In fact, Katz et al. (2017) show that the exact verification task for the ReLU classifiers is NP-complete. Therefore, constructing adversarial attacks and measuring the perturbation's magnitude, which is required to change the prediction, is still one of the most popular ways to estimate the network's robustness. The farther away the adversarial point is from the initial sample, the more robust behavior we expect from the network around this point. Unfortunately, the distance to an adversarial point provides only an upper bound on the distance to the decision boundary. Conversely, formal verification methods output a lower bound on that value by certifying a region around the sample as adversarial-free.

In this work, we develop a novel inexact robustness assessment method that utilizes the fact that the robustness of a network against adversarial perturbations around a given sample is a local property. That is, it does not depend on the behavior of the network outside of the sample's neighborhood. In other words, two networks with similar decision boundaries around the same anchor point must have similar robustness properties, despite showing completely different behavior away from its local neighborhood. Based on these observations, we

1. **develop a novel method to assess the robustness of the deep neural classifiers** based on the local nature of the robustness properties,

2. **develop three pruning techniques** that remove the non-linear activation functions to reduce the complexity of the verification task, but preserve the local behaviour of the network as much as possible: one replacing activations by a constant value, one preserving the output of the initial network for one adversarial point and replacing activation functions by the linear ones, and one preserving the network behaviour in a whole region around the anchor point,

3. **develop a method to identify the neurons to prune** that is based on a bound propagation technique and

4. **empirically verify that the robustness metrics computed on the pruned substitute networks are good estimates of the robustness of the initial network** by conducting experiments on MNIST and CIFAR10 as well as convolutional networks of different sizes.

In Figure 1, we show an example of the difference in the decision boundary before and after we apply one of the proposed pruning techniques in the two-dimensional setting. We remove up to all (1d and 1h) of the hidden neurons while retaining the essential properties of the decision boundary locally around the anchor point. That means to solve the complex task of finding the distance to the decision boundary for the initial model, we save cost by working with the simple local substitute and still get a good approximation of the exact solution.

This work is organized as follows. In Section 2, we introduce the necessary notation and formalize the context of our analysis. In Section 3, we develop both pruning techniques and explain how we focus on the local neighborhood around the base sample. Further, in Section 4, we set up the experimentation workflow and show the results. In Section 5, we mention the relevant related work, and, finally, we present options for future research in Section 6.

## 2 NOTATION

The general idea and our pruning methods apply to any deep classifier, and the constraints apply depending on the deployed attack, verification, and bound propagation approaches. However, to allow for a more straightforward comparison with the existing attacks and verification techniques, we develop our analysis for the classification networks that Li et al. (2023) use for their comprehensive overview and toolbox of robustness verification approaches.

We consider a neural network $f$ consisting of $L$ linear layers, dense or convolutional, and $L-1$ activation layers. We denote the number of neurons in each layer $l$ as $n_l$ for $l \in \{0, \ldots, L\}$, where the

zero's layer contains the input neurons. Furthermore, we consider the $L^\infty$-norm as the perturbation norm since most of the verification methods support it. For a correctly classified anchor point $\boldsymbol{x}^0 \in \mathbb{R}^{n_0}$, weight matrices $\boldsymbol{W}^l \in \mathbb{R}^{n_l \times n_{l-1}}$, and bias vectors $\boldsymbol{b}^l \in \mathbb{R}^{n_l}$, the output of the $i$-th neuron in the $l$-th layer before applying the activation function $\sigma$ is $f_i^l(\boldsymbol{x}^0) = \boldsymbol{W}_{i,:}^l \boldsymbol{x}^{l-1} + \boldsymbol{b}_i^l$ and for the last layer $f^L(\boldsymbol{x}^0) = \boldsymbol{W}^L \boldsymbol{x}^{L-1} + \boldsymbol{b}^L = f(\boldsymbol{x}^0)$. Next, we introduce the notation for four metrics commonly used to assess the robustness of the given classifier. All these algorithms take the network's parameters $\{\boldsymbol{W}^l, \boldsymbol{b}^l\}_{l \in \{1,...,L\}}$ and $\boldsymbol{x}^0$ as input.

**Pre-activation bounds propagation** A pre-activation bounds propagation (of just bounds propagation) algorithm $\mathcal{B}$ is an algorithm that additionally takes a perturbation budget $\epsilon$ as an input and returns a set of the lower and upper bounds $\{\underline{\boldsymbol{a}}^l, \bar{\boldsymbol{a}}^l\}_{l \in \{1,...,L\}}$ on the possible pre-activation values $\boldsymbol{W}^l \boldsymbol{x}^{l-1} + \boldsymbol{b}^l$. In other words, for all possible points $\tilde{\boldsymbol{x}}^0$ with $\|\boldsymbol{x}^0 - \tilde{\boldsymbol{x}}^0\|_\infty \leq \epsilon$ it must hold that $\underline{\boldsymbol{a}}^l \leq \boldsymbol{W}^l \tilde{\boldsymbol{x}}^{l-1} + \boldsymbol{b}^l \leq \bar{\boldsymbol{a}}^l$ in every layer $l$.

**Adversarial attack** An adversarial attack algorithm $\mathcal{A}$ is an algorithm that returns a point $\boldsymbol{x}_{\text{adv}}^0$ that is assigned by $f$ to a different class than $\boldsymbol{x}^0$. We denote the magnitude of the adversarial perturbation $\|\boldsymbol{x}^0 - \boldsymbol{x}_{\text{adv}}^0\|_\infty$ by $\mathcal{A}(f) = d_{\text{adv}}$. The smaller the distance $d_{\text{adv}}$, the better.

**Distance to the decision boundary** Finally, an algorithm $\mathcal{D}$ computing the distance to the decision boundary from $\boldsymbol{x}^0$ is an algorithm that outputs the distance $\|\boldsymbol{x}^0 - \tilde{\boldsymbol{x}}_{\text{adv}}^0\|_\infty$ to the closest adversarial point $\tilde{\boldsymbol{x}}_{\text{adv}}^0$. Formally, $\mathcal{D}$ must solve the following optimization problem.

$$\min_{\tilde{y} \neq y} \min_{\tilde{\boldsymbol{x}}^0 \in \mathbb{R}^{n_0}} \|\tilde{\boldsymbol{x}}^0 - \boldsymbol{x}^0\|_\infty \text{ s.t. } f_y(\boldsymbol{x}^0) \leq f_{\tilde{y}}(\tilde{\boldsymbol{x}}^0), \tag{1}$$

where $y$ is the correctly predicted label of $\boldsymbol{x}^0$. We denote the optimal objective function value of the minimization problem 1 by $\mathcal{D}(f) = d_{\text{bnd}}$.

Note, that for any $\mathcal{A}$ and $\mathcal{D}$ it holds that $d_{\text{bnd}} \leq d_{\text{adv}}$. When we consider a pair of networks $f$ and the pruned network $g$ we define $\hat{d}_{\text{adv}}$ and $\hat{d}_{\text{bnd}}$ as $\mathcal{A}(g)$ and $\mathcal{D}(g)$ correspondingly.

## 3 LOCAL SUBSTITUTES

Given the initial deep classifier, we must reduce its size to construct a neural network that allows for efficient robustness verification. Practitioners pursue the same goal when the network has to be deployed in a setting with strictly constrained available memory or when the forward pass during inference must be particularly fast. In these scenarios, it is common to prune the network by removing its neurons and weights while keeping the drop in performance on the main task as low as possible and even improving generalization to the new data.

However, these available pruning techniques are not suitable for our application. First, all pruning techniques act globally and consider the network's behavior on the whole available dataset. Instead, we show how to prune the network such that only the vicinity of a particular point is considered, which determines its robustness properties. Second, when we assess the robustness of an already trained network, we do not care about the classification error of the resulting network. After reducing the network complexity, our most crucial goal is to preserve a similar decision boundary in a small region around the anchor point. See Section 5 for more information about the available pruning techniques.

To achieve the primary goal - faster verification- we must reduce the number of non-linear transformations within the neural network. The non-linear nature of the neural networks and the large number of non-linear activation functions in their hidden layers are the main reasons deep networks can perform well on complex, high-dimensional data by learning meaningful representations. On the other hand, these non-linear transformations make it hard to explain and verify other properties of the network, like robustness. While the exact reasons for that vary depending on the verification approach, a common problem is to formally describe the output sets of the points from the neighborhood $\mathbb{B}_\epsilon(\boldsymbol{x}^0)$. The more non-linear transformation this set undergoes, the more complex the output becomes and the worse the possible approximations.

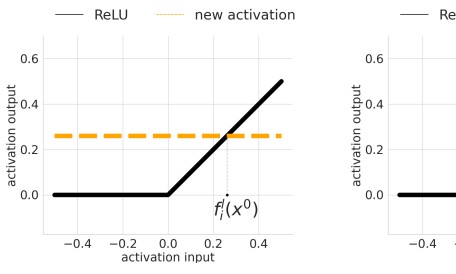 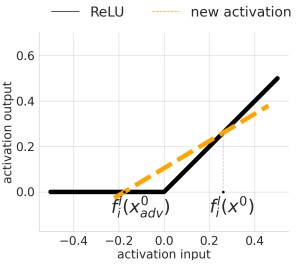 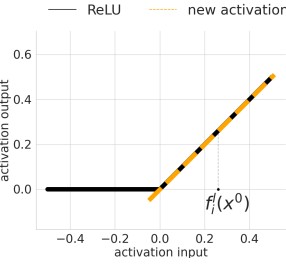

(a) Constant activation function     (b) Linear activation function     (c) Identity activation function

Figure 2: Three options we use to eliminate the non-linear activation function and the corresponding neuron. Figure 2a shows the constant function that always outputs the value $f_i^l(\boldsymbol{x}^0)$. Figure 2b shows the linear function defined by preserving the neuron's output for two points $\boldsymbol{x}^0$ and $\boldsymbol{x}_{\text{adv}}^0$. Finally, Figure 2c shows the function that is either the identity or zero depending on the state of the neuron at $\boldsymbol{x}^0$ (since $f_i^l(\boldsymbol{x}^0)$ is positive in this example, we use the identity function).

### 3.1 How to prune the neurons

For our task, we remove the non-linear transformations from the network $f$ by replacing them with linear or constant functions and getting a new network $g$. We use one of the following three options to retain the network's output at $\boldsymbol{x}_{\text{adv}}^0$ and its neighborhood. For each of the pruning methods, we provide proven evidence that they preserve the network's output values locally around $x^0$ laying theoretical foundation for the empirical evaluations.

**Constant substitute** for the non-linear activation is the simplest option to reduce the complexity of the verification problem $\mathcal{D}$. For each neuron $i$ in layer $l$, we set its output value to $f_i^l(\boldsymbol{x}^0)$, the same as when the anchor point $\boldsymbol{x}^0$ is propagated through the network. This way we ensure that $g(\boldsymbol{x}^0) = f(\boldsymbol{x}^0)$ remains the same. Figure 2a shows how the activation function changes in the example of a ReLU neuron.

One problem that seems to be unavoidable for this type of pruning is what we call boundary collapse. When we prune too much, the resulting network $g$ becomes not expressive enough to maintain a non-trivial boundary even locally around $\boldsymbol{x}^0$. In this case, the prediction region grows, and the classifier outputs the same label for every point, becoming useless for the robustness assessment of the initial network. However, we develop two different pruning techniques to overcome this problem and ensure that even under severe simplification of the decision boundary, it does not collapse but remains near $\boldsymbol{x}^0$ as follows.

**Adversarial preserving linear substitute** uses an adversarial attack $\mathcal{A}$ to determine the radius $d_{\text{adv}}$ of the relevant neighborhood around $\boldsymbol{x}^0$. Now, we do not only ensure that the network's output at the base point $\boldsymbol{x}^0$ remains the same but also at the point $\boldsymbol{x}_{\text{adv}}^0$ that is assigned to a different class. That means we force $g$ to output the same value as $f$ in the center of $\mathbb{B}_{d_{\text{adv}}}(\boldsymbol{x}^0)$ and at one point on its boundary. Therefore, by maintaining $g$'s predicted label at these points, we implicitly enforce that the decision boundary does not move farther away than $\boldsymbol{x}_{\text{adv}}^0$.

To achieve this adversarial preserving property, we replace the non-linear activation function with a linear function with the same output for the points $\boldsymbol{x}^0$ and $\boldsymbol{x}_{\text{adv}}^0$. Figure 2b shows an example of the new activation function for a ReLU neuron. Note that there is always precisely one linear function that satisfies this property and goes through the points $(f_i^l(\boldsymbol{x}^0), \sigma(f_i^l(\boldsymbol{x}^0)))$ and $(f_i^l(\boldsymbol{x}_{\text{adv}}^0), \sigma(f_i^l(\boldsymbol{x}_{\text{adv}}^0)))$ unless $f_i^l(\boldsymbol{x}^0) = f_i^l(\boldsymbol{x}_{\text{adv}}^0)$, in which case we set the new activation function to be constant as we do it above for the bounds propagation based pruner.

**Activation pattern preserving linear substitute** is the third option for pruning the ReLU networks with an even stronger similarity property between $f$ and $g$. If a neuron is activated at $\boldsymbol{x}^0$, we replace the ReLU function with the identity and otherwise by the constant zero function. Due to the continuity of the ReLU, the new network $g$ has the same output as $f$ not only in a single point but in a neighborhood around $\boldsymbol{x}^0$. That means, there exists $\delta > 0$ such that $f(\tilde{\boldsymbol{x}}^0) = g(\tilde{\boldsymbol{x}}^0)$ for all $\tilde{\boldsymbol{x}}^0 \in B_\delta(\boldsymbol{x}^0)$. This neighborhood is defined as the set of points with the same activation pattern for the pruned neurons as $\boldsymbol{x}^0$. However, the more neurons we prune, the smaller this set becomes. See

Appendix B for further details, including a proven result that directly links the robustness verification problem of $f$ and $g$. Figure 2b shows the new activation function for the case $f_i^l(\boldsymbol{x}^0) > 0$.

## 3.2 How to choose the neurons to prune

Bounds propagation techniques, as defined in Section 2, are widely used in verification literature. The most straightforward way to utilize them for the ReLU networks is to detect the neurons that do not change their state for the inputs from $\mathbb{B}_\epsilon(\boldsymbol{x}^0)$. If $\bar{a}_i^l \leq 0$, the input to the neuron $i$ from layer $l$ is always negative or zero, and the neuron remains deactivated such that the activation function returns zero for all relevant inputs. Also, if we get the case $\underline{a}_i^l \geq 0$, this neuron is never deactivated and effectively acts as the identity function. By pruning these stable neurons, we get the new smaller network $g$ with precisely the same output $g(\tilde{\boldsymbol{x}}^0) = f(\tilde{\boldsymbol{x}}^0)$ for all $\tilde{\boldsymbol{x}}^0 \in \mathbb{B}_\epsilon(\boldsymbol{x}^0)$.

Unfortunately, this does not work for the not piece-wise linear activation functions like sigmoid. Moreover, in this procedure, the only parameter that controls the number of the removed non-linear transformations is $\epsilon$, which defines the size of $\mathbb{B}_\epsilon(\boldsymbol{x}^0)$. The smaller the input region around $\boldsymbol{x}^0$, the tighter are the bounds $\underline{a}_i^l, \bar{a}_i^l$, such that less of the neurons remain in the undecided state $\underline{a}_i^l \leq 0 \leq \bar{a}_i^l$. However, we use this parameter to specify the region we consider for pruning. When $\epsilon$ is chosen too small to remove more neurons, we risk that the decision boundary around $\boldsymbol{x}^0$ will lie entirely outside of $\mathbb{B}_\epsilon(\boldsymbol{x}^0)$, meaning that the whole set will be assigned to the same class as $\boldsymbol{x}^0$. This is not desirable since the computation of the bounds in $\mathcal{B}$ does not account for the behavior of the network at the points outside of the given $\epsilon$-neighborhood around $\boldsymbol{x}^0$, which, in this case, does determine the network's robustness properties. That means we would miss essential information about the network when computing the bounds $\underline{a}_i^l, \bar{a}_i^l$ in the first place.

Instead, we first set $\epsilon = d_{\text{adv}}$ by running an attack $\mathcal{A}$. This way, we ensure that $\mathbb{B}_\epsilon(\boldsymbol{x}^0)$ is not too small and contains the decision boundary between the points with different predicted classes $\boldsymbol{x}^0$ and $\boldsymbol{x}_{\text{adv}}^0$ (both are points from $\mathbb{B}_\epsilon(\boldsymbol{x}^0)$). Given a user-defined ratio $\gamma \in (0, 1)$ of the neurons to be pruned, we run the bounds propagation algorithm $\mathcal{B}$ and remove the neurons with the smallest difference $\bar{a}_i^l - \underline{a}_i^l$, that we use as a measure of the neuron's variability given the input perturbation budget $d_{\text{adv}}$. That is, we prune the neurons with the least variable output, considering the inputs from $\mathbb{B}_{d_{\text{adv}}}(\boldsymbol{x}^0)$.

Finally, we observe that the neurons of the first layer have $\bar{a}_i^1$ and $\underline{a}_i^1$ with the smallest gap due to the magnification during bounds estimation when $\bar{a}_i^l$ and $\underline{a}_i^l$ are propagated in $\mathcal{B}$ and become looser. Therefore, to prevent the elimination of the neurons exclusively from the first layers, we apply the pruning iteratively by removing a small portion at a time and then recomputing the pre-activation bounds. This way, each time, we prune neurons with the most tight bounds and distribute the removed neurons more evenly throughout the layers. We present the described method in Algorithm 1.

---

**Algorithm 1:** Bounds propagation based iterative pruning

1: **Input:** Algorithms $\mathcal{A}, \mathcal{B}$, network $f$, point $\boldsymbol{x} \in \mathbb{R}^{n_0}$, $\gamma \in (0, 1)$ and $\gamma_0 \ll \gamma$.
2: **Output:** Network with at most $(1 - \gamma)(n_1 + \cdots + n_L)$ ReLU activations.
3: Compute $d_{\text{adv}} = \mathcal{A}(f)$.
4: Compute $\bar{a}^l, \underline{a}^l$ using $\mathcal{B}$ with $\epsilon = d_{\text{adv}}$, initialize $k = 1$.
5: **repeat**
6:     Compute the threshold $\delta$ as the $\gamma_0$-quantile of the values $\bar{a}_i^l - \underline{a}_i^l$ for $l \in \{2, \ldots, L\}$ and $i \in \{1, \ldots, n_l\}$.
7:     **for** $l = 2, \ldots, L$ **do**
8:         Define $I^l \in \{\text{True}, \text{False}\}^{n_l}$ as the indicator vector of the remaining neurons: $I_i^l = \mathbb{1}_{\bar{a}_i^l - \underline{a}_i^l > \delta}$
9:         For $i \in \{1, \ldots, n_l\}$ where $I_i^l$ is False replace ReLU by a linear function according to Section 3.1.
10:     **end for**
11:     Update $k = k + 1$.
12: **until** $(1 - \gamma_0)^k \geq 1 - \gamma$
13: **Return:** Network with the new activation layers.

---

By applying our pruning methods, we achieve the following. 1. **We preserve local behavior** of the network around $x^0$ by pruning the least uncertain neurons, while we don't take into account the points outside of $\mathbb{B}_{d_{\text{adv}}}(x^0)$. 2. **We preserve the value of the network either at $x^0$, at $x^0$ and $x^0_{\text{adv}}$, or even in a neighborhood of $x^0$**, by construction of $g$ as described in Section 3.1. 3. **We have a dedicated parameter $\gamma$ to set the desired amount of the pruned neurons**.

## 4 EXPERIMENTS

**Setup**  We conduct the experiments using three convolutional architectures with ReLU activations publicly provided by Li et al. (2023) that are pretrained on the MNIST and CIFAR10 datasets. We call them according to authors' notation C–E, where C is the smallest network with 166406 parameters, and E is the largest with 2466858 parameters. The models are trained either normally or using adversarial training. The following perturbation magnitudes are used to generate adversarial examples during adversarial training: 0.1 or 0.3 for MNIST, 0.008 or 0.031 for CIFAR10. We evaluate the verification time of all approaches on a machine with a single NVIDIA GTX 1080 Ti GPU with 11 GB memory. For the algorithms $\mathcal{A}$, $\mathcal{B}$, and $\mathcal{D}$, we use the following popular approaches correspondingly (all available within our implementation based on the unified toolbox by Li et al. (2023) that will be publicly available): projected gradient descent (PGD) attack, FastLinIBP interval bound propagation (combination of the methods by Weng et al. (2018) and Gowal et al. (2018)), MILP-based exact verification by Tjeng et al. (2018). Unless we mention otherwise, all third-party methods are used with the default parameters by Li et al. (2023).

**Procedure**  The goal is to get a good estimate of $d_{\text{bnd}}$ without running $\mathcal{D}$ on the full network. Therefore, we apply the three pruning methods (Section 3.1) with varying magnitudes for each network. To choose the neurons to prune, we use the bounds propagation-based approach (Algorithm 1) with $\gamma \in \{0.9, 0.99\}$. Note, that the pruning step is independent of the downstream task solved for the pruned model. The same holds for the key insight we verify in our work: the proposed pruning procedure **preserves the network's local boundary** around the anchor point.

Before and after the pruning, we run $\mathcal{D}$ on 10% of the test samples. We use a particular complete verification approach in our experiments, but the core property of the pruning method (i.e., preserving the boundary) does not depend on it. Furthermore, for other complete verification approaches, the resulting $d_{\text{bnd}}$ and $\hat{d}_{\text{bnd}}$ must be the same since the verification approaches are complete and solve the same task. Thus, the difference in the computed radii does not depend on choosing a particular complete verification approach. For all the pruned networks, we compute the mean absolute difference $|\hat{d}_{\text{bnd}} - d_{\text{bnd}}|/d_{\text{bnd}}$. The goal is to construct an estimate close to $d_{\text{bnd}}$ and do it faster than $\mathcal{D}$ when applied to the initial model $f$. However, due to the complexity of $\mathcal{D}$, the exact verification often terminates without reaching the optimal point of the MILP optimization problem due to the maximum runtime constraint for a single optimization run. Therefore, to compute the difference in the distance to the decision boundaries fairly, we consider only the points where the algorithm $\mathcal{D}$ does compute the true $\hat{d}_{\text{bnd}}$ and $d_{\text{bnd}}$ (see Appendix B for more details).

The primary instance of the MILP-based verification approach by Tjeng et al. (2018) is the mixed-integer optimization problem that must be solved for each sample, target label, and the currently verified radius. It is applied within a binary search to find $d_{\text{bnd}}$ as the largest radius where a successful robustness verification is possible. We compare the average runtime of the solver needed to solve this task before and after pruning. However, we have to take into account the fact that the larger the verified radius $\epsilon$ is, the more complex the corresponding task becomes as the set of admissible points $\mathbb{B}_{\epsilon}(x^0)$ becomes larger. As the pruned models have fewer non-linear functions and fewer binary variables in the MILP formulation, we verify larger radii for these models. Therefore, we restrict the set of the solver runs used to compute the average runtime to those that evaluate the robustness of the pruned models for the radii from the same range as for the unpruned models. Thus, we ensure that the computed average runtime represents the solver runs solving the verification task of the same complexity with respect to $\epsilon$. We provide more details about this issue in Appendix B.

**Results**  Table 1 shows the results for the architecture C. For the complete collection of the results, please see Appendix C. Each row represents an evaluation of an initial and pruned network pair. Training methods *clean*, *adv1*, and *adv2* refer to the standard and adversarial training procedures

Table 1: Results for the architecture C.

| Network Dataset / Trn | Pruning Type / $\gamma$ | Avg $d_{\text{bnd}}$ diff | Avg time (s) pruning div. by #runs | opt. run with pruning | opt. run w/o pruning |
|---|---|---|---|---|---|
| mnist / clean | con / 0.9 | 0.39 ± 0.063 | 0.02 ± 0.01 | 0.73 ± 0.56 | 33.28 ± 26.14 |
| mnist / clean | con / 0.99 | 0.39 ± 0.063 | 0.04 ± 0.02 | 0.39 ± 0.22 | 33.28 ± 26.14 |
| mnist / clean | lin / 0.9 | 0.0 ± 0.0 | 0.04 ± 0.03 | 31.57 ± 24.16 | 33.28 ± 26.14 |
| mnist / clean | lin / 0.99 | 0.0054 ± 0.0062 | 0.06 ± 0.03 | 16.91 ± 17.95 | 33.28 ± 26.14 |
| mnist / clean | act / 0.9 | 0.016 ± 0.011 | 0.04 ± 0.03 | 31.99 ± 24.49 | 33.28 ± 26.14 |
| mnist / clean | act / 0.99 | 0.019 ± 0.01 | 0.06 ± 0.04 | 17.19 ± 18.64 | 33.28 ± 26.14 |
| mnist / adv1 | con / 0.9 | 0.19 ± 0.061 | 0.03 ± 0.02 | 10.91 ± 19.90 | 34.01 ± 25.84 |
| mnist / adv1 | con / 0.99 | 0.72 ± 0.76 | 0.08 ± 0.05 | 0.79 ± 0.55 | 34.01 ± 25.84 |
| mnist / adv1 | lin / 0.9 | 0.0 ± 0.0 | 0.05 ± 0.04 | 33.78 ± 25.62 | 34.01 ± 25.84 |
| mnist / adv1 | lin / 0.99 | 0.0068 ± 0.012 | 0.07 ± 0.04 | 9.73 ± 14.12 | 34.01 ± 25.84 |
| mnist / adv1 | act / 0.9 | 0.0067 ± 0.0058 | 0.03 ± 0.02 | 30.93 ± 26.11 | 34.01 ± 25.84 |
| mnist / adv1 | act / 0.99 | 0.01 ± 0.01 | 0.05 ± 0.02 | 3.75 ± 5.91 | 34.01 ± 25.84 |
| mnist / adv2 | con / 0.9 | 0.061 | 0.04 ± 0.03 | 20.51 ± 26.21 | 33.10 ± 25.96 |
| mnist / adv2 | con / 0.99 | 0.3 | 0.08 ± 0.06 | 0.68 ± 0.56 | 33.10 ± 25.96 |
| mnist / adv2 | lin / 0.9 | 0.0 | 0.06 ± 0.05 | 37.64 ± 24.63 | 33.10 ± 25.96 |
| mnist / adv2 | lin / 0.99 | 0.031 | 0.06 ± 0.06 | 7.38 ± 11.24 | 33.10 ± 25.96 |
| mnist / adv2 | act / 0.9 | 0.01 | 0.03 ± 0.02 | 31.57 ± 26.17 | 33.10 ± 25.96 |
| mnist / adv2 | act / 0.99 | 0.031 | 0.04 ± 0.02 | 2.17 ± 1.94 | 33.10 ± 25.96 |
| cifar10 / clean | con / 0.9 | 2.1 ± 0.83 | 0.02 ± 0.01 | 1.35 ± 1.52 | 60.12 ± 52.64 |
| cifar10 / clean | con / 0.99 | 5.5 ± 5.7 | 0.03 ± 0.02 | 0.24 ± 0.26 | 60.12 ± 52.64 |
| cifar10 / clean | lin / 0.9 | 0.0098 ± 0.014 | 0.02 ± 0.01 | 37.22 ± 48.50 | 60.12 ± 52.64 |
| cifar10 / clean | lin / 0.99 | 0.0098 ± 0.014 | 0.04 ± 0.03 | 9.20 ± 20.13 | 60.12 ± 52.64 |
| cifar10 / clean | act / 0.9 | 0.0098 ± 0.014 | 0.02 ± 0.02 | 39.01 ± 48.82 | 60.12 ± 52.64 |
| cifar10 / clean | act / 0.99 | 0.0098 ± 0.014 | 0.04 ± 0.03 | 11.01 ± 21.63 | 60.12 ± 52.64 |
| cifar10 / adv1 | con / 0.9 | 2.8 ± 0.21 | 0.01 ± 0.01 | 1.44 ± 5.23 | 63.00 ± 54.41 |
| cifar10 / adv1 | con / 0.99 | 47 ± 69 | 0.04 ± 0.02 | 0.62 ± 0.33 | 63.00 ± 54.41 |
| cifar10 / adv1 | lin / 0.9 | 0.019 ± 0.032 | 0.03 ± 0.02 | 47.08 ± 50.26 | 63.00 ± 54.41 |
| cifar10 / adv1 | lin / 0.99 | 0.019 ± 0.032 | 0.06 ± 0.04 | 19.27 ± 31.93 | 63.00 ± 54.41 |
| cifar10 / adv1 | act / 0.9 | 0.0 ± 0.0 | 0.03 ± 0.02 | 44.74 ± 49.35 | 63.00 ± 54.41 |
| cifar10 / adv1 | act / 0.99 | 0.0 ± 0.0 | 0.04 ± 0.02 | 14.89 ± 29.26 | 63.00 ± 54.41 |
| cifar10 / adv2 | con / 0.9 | 1.5 ± 0.43 | 0.02 ± 0.01 | 10.77 ± 29.97 | 59.76 ± 55.19 |
| cifar10 / adv2 | con / 0.99 | 3.3 ± 2.6 | 0.04 ± 0.02 | 0.69 ± 0.41 | 59.76 ± 55.19 |
| cifar10 / adv2 | lin / 0.9 | 0.013 ± 0.025 | 0.04 ± 0.04 | 49.03 ± 51.70 | 59.76 ± 55.19 |
| cifar10 / adv2 | lin / 0.99 | 0.017 ± 0.026 | 0.06 ± 0.05 | 22.78 ± 38.50 | 59.76 ± 55.19 |
| cifar10 / adv2 | act / 0.9 | 0.0018 ± 0.0055 | 0.03 ± 0.02 | 46.50 ± 50.80 | 59.76 ± 55.19 |
| cifar10 / adv2 | act / 0.99 | 0.0033 ± 0.0071 | 0.04 ± 0.01 | 13.76 ± 30.00 | 59.76 ± 55.19 |

with different attack budgets, as mentioned above. We encode the pruning approach by one of the three options *constant*, *linear* or *activation* described in Section 3.1 and Figures 2a-2c as well as the sparsity level $\gamma$. Column *Avg $d_{bnd}$ diff* contains the mean difference of the computed distance to the decision boundary without reaching the time limit. Column *Avg. time (s) pruning div. by #runs* shows the time needed to obtain the pruned model divided by the number of optimization runs (for different target labels and verified radii) for this model, averaged over all anchor points. Columns *Avg time with pruning* and *Avg time w/o pruning* show the average time needed to solve a single verification MILP after and before pruning for similar $\epsilon$ as we described above. In addition to these metrics, we provide the corresponding standard deviations after the $\pm$ sign (the standard deviation of *Avg bnd diff* for the model *mnist / adv2* are missing since $\mathcal{D}$ terminates without exceeding the runtime limit only for a single datapoint).

Results clearly show that the proposed method achieves a consistent speedup of the verification procedure. When 99% of the ReLU functions are removed, we solve the atomic mixed-integer tasks 2 to 5 times faster. The constant pruning achieves the most speedup as it allows not only to reduce

the number of the binary variables that encode the state of the ReLU but the continuous variables corresponding to the pruned neurons as well (see Appendix B).

Note, that the longest average pruning time is just 0.325s, spent for model E, CIFAR10 dataset with adversarial preserving linear pruning and 99% sparsity (compared to the average MILP solution time of over a minute). It shows that the pruning step applies to larger models without significant runtime overhead compared to the complete verification procedure.

Looking at the distance difference to the decision boundary before and after the pruning, we report that the adversarial and activation pattern preserving linear ReLU-substitutes result in a local behavior of $g$ similar to $f$. Due to their theoretical properties, these pruning methods keep the decision boundary around $x^0$, such that the average relative absolute difference between $\hat{d}_{\text{bnd}}$ and $d_{\text{bnd}}$ stays at around 1%. Thus, we empirically justify the usage of the smaller substitute models constructed by considering the initial network's local properties for the robustness verification task. Contrary to these two pruning methods, the *constant* pruning does not lead to any useful approximations of $d_{\text{bnd}}$ as the relative difference is 100% or higher for most models. Overall, the adversarial preserving pruner provides an excellent tool for practitioners not interested in the exact verification of their network but need a faster way to estimate the distance to the nearest adversarial point.

## 5 RELATED WORK

Regarding robustness verification, two prominent families of methods are used in practice: complete (or exact) and incomplete. **Complete verification** techniques verify exactly whether an adversarial example exists inside the given perturbation budget. The theoretical formulation of this direction uses the mixed integer linear programming (MILP) framework, for which multiple solvers are available based on either satisfiability modulo theory (SMT) Katz et al. (2017) or integer programming methods (Tjeng & Tedrake, 2017; Dutta et al., 2017).

**Incomplete verification** techniques relax the constraints on the activations of each layer in the neural network. An example is the work by Wong & Kolter (2018), where the authors replace ReLU activations with a convex substitute and, to increase the method's efficiency further, use the dual formulation of the obtained relaxed linear programming (LP) problem. Ehlers (2017) use a similar approximation of the ReLU. However, they do not optimize over the dual LP and solve the problem using SMT. Following this direction Weng et al. (2018); Zhang et al. (2018) approximate the ReLU activation functions and propagate the possible bounds of the input to the final layer. Other approaches use semi-definite programming (Raghunathan et al., 2018a;b), Lipschitz constant (Weng et al., 2018; Hein & Andriushchenko, 2017; Zhang et al., 2019; Tsuzuku et al., 2018) and abstraction of the network (Gehr et al., 2018). Similar to our work, Croce et al. (2019) look at the local area around the base point and, in particular, utilize the fact that (in the case of ReLU activations) it consists of regions where the network act as a linear function. Besides their incomplete verification method, the authors propose a robust training loss.

All incomplete verification approaches provide a lower bound on the distance to the decision boundary, which is strongly dependent on the tightness of the used relaxation and how each algorithm computes the bounds on the pre-activation values in each layer. Li et al. (2023) provide a detailed overview of the verification methods, including a proposed taxonomy and a publicly available verification toolbox.

Pruning methods allow for minimizing the models' size at only a small cost of their accuracy. They eliminate weights, neurons, or layers so the output changes as little as possible. Hoefler et al. (2021) divide the pruning methods into two categories: **data independent pruning** and **data dependent pruning**. The first group consists of pruning methods that do not explicitly use the training data. These methods consider the weights and nodes of the network after training and heuristically find a maximal subset of them that have the minimal impact on the final accuracy (Li et al., 2016; Han et al., 2015; Jonathan & Michael, 2019). Shumailov et al. (2019) consider the transferability of adversarial examples between the pruned (Han et al., 2015) and initial model. Their findings are connected to our work since we also deal with the transferability of the robustness properties to the network after pruning. However, the authors suggest that adversarial examples are interchangeable only at low sparsity levels, which, unfortunately, limits the potential gain in computational cost.

**Data dependent pruning** methods consider the effect of the training data on the network's output when deciding which neurons or weights to prune. Specifically, these methods consider the influence of the training data on either the output, the activations, or the network's derivatives and eliminate the network's extraneous parts without using its weights and biases directly. This idea does not apply well to our scenario, as the training data can not be used for pruning while preserving the local decision boundary. We mention them as they are successful in accuracy-preserving pruning (Lee et al., 2019; Wang et al., 2020; Evci et al., 2020).

Finally, another method to create a small network with similar robustness properties as the initial large network is to **train it from scratch**. For example, using the initial network's output to guide the training process. Papernot et al. (2016) use this approach to create a substitute network for attacking the initial model in a black-box environment (that means without access to its gradients). They specifically train the substitute to mimic the initial network on carefully selected data points, iteratively picked by going towards the decision boundary using the gradient information of the substitute network. This method seems very promising, as they also look at the problem by creating a network with a similar decision boundary to the initial one. However, we abstain from performing training in our use case since we must do it for each of the considered base points (unlike Papernot et al. (2016), who train a single substitute network) that is potentially even slower than the verification on $f$.

## 6 Conclusion

Successful sustainable integration of machine learning methods into areas relevant to the industry and society is a challenging task. Especially after deep learning based technologies have proven to be able to tackle complex classification problems, we have to be cautious. We should not rely on the demonstrated excellent performance only.

The presented verification method contributes to the general acceptance of deep learning models. We develop a novel framework to estimate the robustness of a deep classifier around a given base point. We apply the available MILP-based method the exact robustness verification to a specifically constructed local substitute network. We observe that the decision boundary of the initial network far away from the base point does not affect its robustness against adversarial perturbations in the local neighborhood and develop three pruning techniques that preserve the local properties of the network while reducing its overall complexity. This idea allows for a more efficient application of the verification method $\mathcal{D}$ on the pruned models. At the same time, the resulting metrics are reasonable estimates for the robustness of the initial network because of the similarity in the local behavior. We conduct the experiments on the MNIST and CIFAR10 datasets and a variety of convolutional networks. The results show that, in particular, the adversarial and activation pattern preserving pruners (see Section 3.2) approximate $d_{\text{bnd}}$ very well (around 1% absolute relative difference, when at least 90% of activations are removed) with a significant speedup compared to $\mathcal{D}$ applied on the initial model. In summary, the proposed technique allows practitioners interested in faster inexact robustness assessment to get a reliable evaluation of the robustness properties.

**Future work** (Katz et al., 2017) show that the $\epsilon$-verification task and thus also computing the distance to the decision boundary is NP-complete. Therefore, we do not set the goal to generally make complete verification computationally tractable, as it is impossible due to the complexity of the underlying optimization task. Instead, we believe that providing reliable estimates of the robustness of the network is a very promising direction.

The success of the presented approach indicates an intriguing direction for future work toward larger networks and datasets. Also, conducting the experiments with different $\mathcal{D}$ should verify that the speedup we get is agnostic to the applied verification methods. Furthermore, we think about deriving additional theoretical guarantees about the difference of $d_{\text{bnd}}$ and $\hat{d}_{\text{bnd}}$. See Appendix A for the result of this type and other findings.

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
