# A   THEORETICAL GUARANTIES

We show that the local similarity of two networks allows us to derive robustness guarantees for one network if we prove similar properties for the second one without running the verification algorithm a second time. Here, we address the following so-called $\epsilon$-verification problem. Like in Section 2, given the anchor point $\boldsymbol{x}^0$, network $f$, predicted label $y$, and a target label $\hat{y}$, we consider the following problem, which, in case of the ReLU activation, is equivalent to a mixed-integer optimization task.

$$\min_{\tilde{\boldsymbol{x}}^0 \in \mathbb{R}^{n_0}} f_y(\tilde{\boldsymbol{x}}^0) - f_{\hat{y}}(\tilde{\boldsymbol{x}}^0) \text{ s.t. } \|\boldsymbol{x}^0 - \tilde{\boldsymbol{x}}^0\| \leq \epsilon \tag{2}$$

We denote its optimal value by $\mathcal{V}_f(\epsilon)$. Note, that $\mathcal{V}_f(\epsilon) \geq 0 \iff f_y(\tilde{\boldsymbol{x}}^0) \geq f_{\hat{y}}(\tilde{\boldsymbol{x}}^0)$ for all $\tilde{\boldsymbol{x}}^0 \in \mathcal{B}_\epsilon(\boldsymbol{x}^0)$, meaning that the initial label $y$ has a higher score than $\hat{y}$ everywhere on $\mathcal{B}_\epsilon(\boldsymbol{x}^0)$. In the MILP-based algorithms $\mathcal{D}$ this problem is solved for different $\epsilon$ to find the largest $\epsilon$, such that $\mathcal{V}_f(\epsilon) \geq 0$, which is exactly the distance to the decision boundary $d_{bnd}$.

**Lemma 1.** *If $f(\tilde{\boldsymbol{x}}^0) = g(\tilde{\boldsymbol{x}}^0)$ for all $\tilde{\boldsymbol{x}}^0 \in \mathcal{B}_\delta(\boldsymbol{x}^0)$ and*

1. *$\delta \geq \epsilon$, then $\mathcal{V}_g(\epsilon) \geq 0 \implies \mathcal{V}_f(\epsilon) \geq 0$,*

2. *$\delta \leq \epsilon$, then $\mathcal{V}_g(\epsilon) \geq 2(L_f + L_g)(\epsilon - \delta) \implies \mathcal{V}_f(\epsilon) \geq 0$,*

*where $L$ is the Lipschitz constant of the corresponding function on $\mathcal{B}_\epsilon(\boldsymbol{x}^0)$.*

*Proof.* 1 follows directly from the fact that $f = g$ on the set $\mathcal{B}_\delta(\boldsymbol{x}^0)$ that includes the admissible set of the optimization task 2, which is $\mathcal{B}_\epsilon(\boldsymbol{x}^0)$. To prove 2, assume $\boldsymbol{z}^0 \in \mathcal{B}_\epsilon(\boldsymbol{x}^0)$ and let $\tilde{\boldsymbol{z}}^0 = \frac{\epsilon}{\delta}\boldsymbol{z}^0$ be the projection of $\boldsymbol{z}^0$ on $\mathcal{B}_\delta(\boldsymbol{x}^0)$, then

$$f_y(\boldsymbol{z}^0) - f_{\hat{y}}(\boldsymbol{z}^0) = \underbrace{f_y(\boldsymbol{z}^0) - f_y(\tilde{\boldsymbol{z}}^0)}_{\geq -L_f\|\boldsymbol{z}^0-\tilde{\boldsymbol{z}}^0\|} + \underbrace{f_y(\tilde{\boldsymbol{z}}^0) - g_y(\tilde{\boldsymbol{z}}^0)}_{=0} + \underbrace{g_y(\tilde{\boldsymbol{z}}^0) - g_y(\boldsymbol{z}^0)}_{\geq -L_g\|\boldsymbol{z}^0-\tilde{\boldsymbol{z}}^0\|} + \underbrace{g_y(\boldsymbol{z}^0) - g_{\hat{y}}(\boldsymbol{z}^0)}_{\geq \mathcal{V}_g(\epsilon)} +$$
$$+ \underbrace{g_{\hat{y}}(\boldsymbol{z}^0) - g_{\hat{y}}(\tilde{\boldsymbol{z}}^0)}_{\geq -L_g\|\boldsymbol{z}^0-\tilde{\boldsymbol{z}}^0\|} + \underbrace{g_{\hat{y}}(\tilde{\boldsymbol{z}}^0) - f_{\hat{y}}(\tilde{\boldsymbol{z}}^0)}_{=0} + \underbrace{f_{\hat{y}}(\tilde{\boldsymbol{z}}^0) - f_{\hat{y}}(\boldsymbol{z}^0)}_{\geq -L_f\|\boldsymbol{z}^0-\tilde{\boldsymbol{z}}^0\|}$$
$$\geq \mathcal{V}_g(\epsilon) - 2(L_f + L_g)(\epsilon - \delta)$$

That means $\mathcal{V}_g(\epsilon) \geq 2(L_f + L_g)(\epsilon - \delta)$ indeed implies $f_y(\tilde{\boldsymbol{x}}^0) \geq f_{\hat{y}}(\tilde{\boldsymbol{x}}^0)$ for all $\tilde{\boldsymbol{x}}^0 \in \mathcal{B}_\epsilon(\boldsymbol{x}^0)$, or in other words $\mathcal{V}_f(\epsilon) \geq 0$. □

In our setting, for the network $g$ being constructed via pruning as described in Section 3, Lemma 1 provides a way to certify the neighborhood $\mathcal{B}_\epsilon(\boldsymbol{x}^0)$ as adversarial free for the network $f$ by solving the optimization problem 2 for a smaller network $g$. Why don't we use this theoretical result to certify $f$ but instead rely on the empirical comparison of $d_{bnd}$ and $\hat{d}_{bnd}$? First, only the third from the three developed pruning methods, called *activation preserving pruning* (Figure 2c), leads to a new network $g$ that has the same output as $f$ in a whole neighborhood around $\boldsymbol{x}^0$. The *constant pruning* and *adversarial preserving linear pruning* (Figure 2a and 2b) preserve the output for single points and do not possess this property. Second, for the *activation preserving pruning* we need to know $\delta$, such that $f(\tilde{\boldsymbol{x}}^0) = g(\tilde{\boldsymbol{x}}^0)$ for $\tilde{\boldsymbol{x}}^0 \in \mathcal{B}_\delta(\boldsymbol{x}^0)$. In other words, we must find a region's radius where the pruned neurons' activation pattern does not change. Unfortunately, the task of computing $\delta$, knowing the neurons that we prune, is of the same complexity as the same verification task formulated by 1 itself. More precisely, $\delta$ equals the optimal value of the following problem.

$$\max_{\tilde{\boldsymbol{x}}^0 \in \mathbb{R}^{n_0}} \|\boldsymbol{x}^0 - \tilde{\boldsymbol{x}}^0\| \text{ s.t. } f_i^l(\tilde{\boldsymbol{x}}^0) \geq 0 \text{ if the pruned neuron } (l,i) \text{ is activated, that is } f_i^l(\boldsymbol{x}^0) \geq 0, \text{ and} \tag{3}$$

$$f_i^l(\tilde{\boldsymbol{x}}^0) \leq 0 \text{ otherwise.} \tag{4}$$

Compared to 1, it has the same objective function and the constraints of the same type. Therefore, we do not use Lemma 1 to derive the robustness certificates for the initial network $f$, as solving the intermediate task of computing $\delta$ is not feasible in this framework. Nevertheless, this result proves an underlying theoretical connection between the robustness properties of the initial and pruned networks and provides a starting point for future research in this intriguing direction.

## B   RUNTIME AND COMPLEXITY

To understand what affects the complexity of the problem 2, we reformulate it as follows (here for a single target label $\tilde{y}$).

$$\min_{\tilde{\boldsymbol{x}}, \tilde{\boldsymbol{z}}} (e_{\tilde{y}} - e_y)^T \left( \boldsymbol{W}^L \tilde{\boldsymbol{x}}^{L-1} + \boldsymbol{b}^L \right), \text{ s.t. } \|\boldsymbol{x}^0 - \tilde{\boldsymbol{x}}^0\| \leq \epsilon \text{ and for } l \in [L-1] \tag{5}$$

$$\tilde{\boldsymbol{x}}^l \leq \boldsymbol{W}^l \tilde{\boldsymbol{x}}^{l-1} + \boldsymbol{b}^l - (\boldsymbol{1} - \boldsymbol{z}^l)\underline{\boldsymbol{a}}^l \tag{6}$$

$$\tilde{\boldsymbol{x}}^l \geq \boldsymbol{W}^l \tilde{\boldsymbol{x}}^{l-1} + \boldsymbol{b}^l \tag{7}$$

$$\tilde{\boldsymbol{x}}^l \leq \boldsymbol{z}^l \bar{\boldsymbol{a}}^l, \ \tilde{\boldsymbol{x}}^l \geq 0 \tag{8}$$

$$x_i^l \in \mathbb{R}, z_i^l \in \{0,1\}$$

We use this formulation in our implementation based on the work by Li et al. (2023). Note, that once we prune a neuron $(l, i)$ as described in Section 3, the corresponding binary variable $z_i^l$ disappears from the formulation together with the constraints 6–8 and we get a simple linear constraint $\tilde{\boldsymbol{x}}_i^l = \boldsymbol{W}_{i,:}^l \tilde{\boldsymbol{x}}^{l-1} + \boldsymbol{b}_i^l$ instead. Here, $\boldsymbol{W}_{i,:}^l$ and $\boldsymbol{b}_i^l$ define our new linear transformation instead of the non-linear ReLU activation. By pruning, for example, 99% of the hidden neurons, we reduce the number of integer variables in 1 by the same amount. The standard approach to solve a MILP task is branch-and-bound, where we divide the whole set of the admissible points by relaxing the integrality constraints (in our case $z_i^l \in \{0,1\}$) one by one and bound the optimal value on the resulting sub-domains. Fewer binary variables mean less branching and a faster problem solution.

On the other hand, smaller $\epsilon$ also contributes to a faster solution. Now, instead of affecting the ReLU-constraints 6–8, $\epsilon$ directly controls the size of $\mathcal{B}_\epsilon(\boldsymbol{x}^0)$ via the constraint $\|\boldsymbol{x}^0 - \tilde{\boldsymbol{x}}^0\| \leq \epsilon$. Again, smaller $\epsilon$ results in a smaller admissible set that gets explored faster by the solver. To demonstrate this correlation, in Figures 3a and 3b, we show the time of each optimization run solving problem 2 that completed within the time limit of 60 seconds on the $x$-axis. The $\epsilon$ that was verified is shown on the $y$-axis. Each point represents the optimization tasks solved on model C pruned either using *linear adversarial preserving* or *activation pattern preserving* method indicated by the color of the points. For Figure 3a we use the sparsity parameter $\gamma = 0.9$ and for Figure 3b $\gamma = 0.99$. That means within one plot, the only parameter affecting the optimization task's complexity is $\epsilon$.

## C   TABLES

Below, Tables 2 and 3 show the results for the architectures D and E, correspondingly. For these larger networks, the MILP-based method $\mathcal{D}$ failed to find the exact $d_{bnd}$ for all the considered MNIST and CIFAR10 samples. In other words, it reaches the time limit of 120 seconds for MNIST and 180 seconds for CIFAR10 for at least one of the optimization tasks 2 solved during the verification process. Therefore, the tables do not contain the column *Avg $d_{bnd}$ diff*, as we do not have the true $d_{bnd}$ and $\hat{d}_{bnd}$ to compare. Furthermore, because of the long computing time for architecture E, we verify the networks trained using adversarial training only (no clean training).

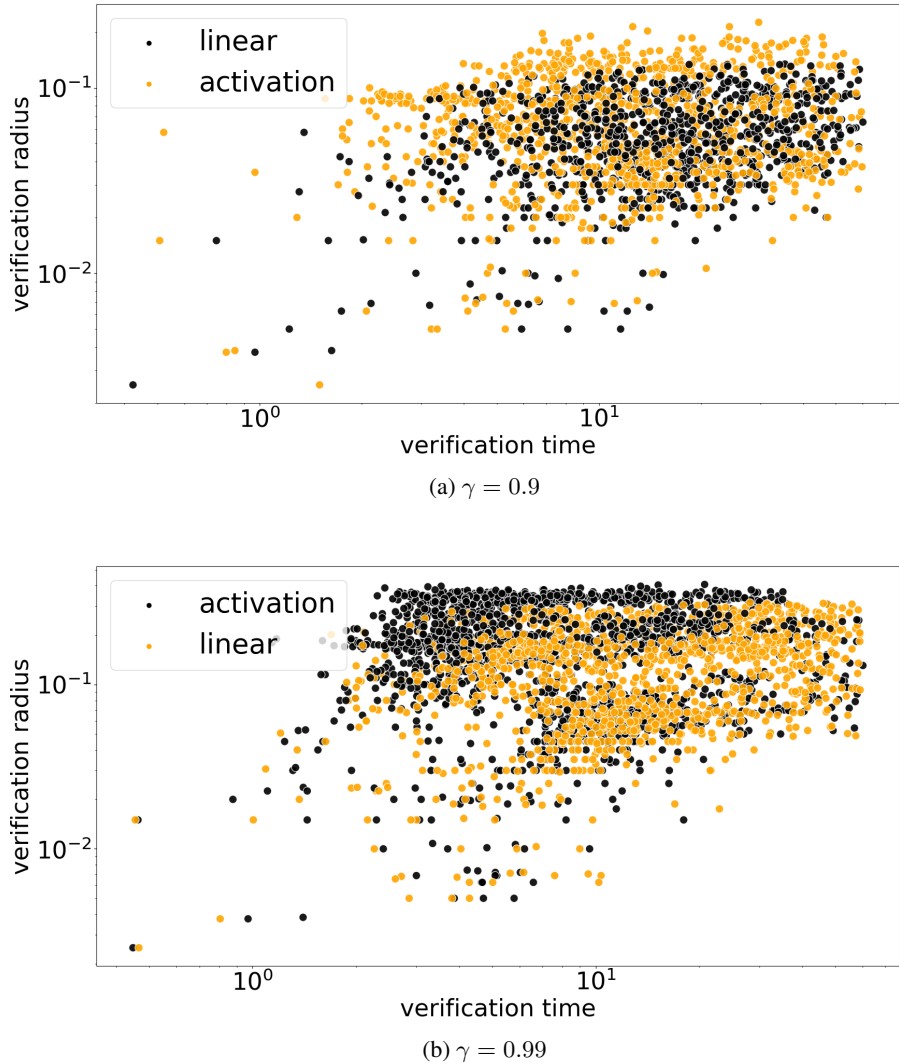

(a) $\gamma = 0.9$

(b) $\gamma = 0.99$

Figure 3: Runtime and verified $\epsilon$ for the optimization tasks for model C.

Table 2: Results for the architecture D.

| Dataset / Training | Pruning | $\gamma$ | Avg time with pruning | Avg time w/o pruning |
|---|---|---|---|---|
| mnist / clean | constant | 0.9 | 32.52 ± 24.94 | 46.71 ± 21.97 |
| mnist / clean | constant | 0.99 | 2.25 ± 0.98 | 46.71 ± 21.97 |
| mnist / clean | linear | 0.9 | 46.89 ± 21.68 | 46.71 ± 21.97 |
| mnist / clean | linear | 0.99 | 45.20 ± 20.89 | 46.71 ± 21.97 |
| mnist / clean | activation | 0.9 | 46.33 ± 22.23 | 46.71 ± 21.97 |
| mnist / clean | activation | 0.99 | 44.86 ± 21.02 | 46.71 ± 21.97 |
| mnist / adv1 | constant | 0.9 | 38.12 ± 23.40 | 47.15 ± 21.26 |
| mnist / adv1 | constant | 0.99 | 3.45 ± 1.94 | 47.15 ± 21.26 |
| mnist / adv1 | linear | 0.9 | 47.66 ± 21.05 | 47.15 ± 21.26 |
| mnist / adv1 | linear | 0.99 | 47.05 ± 20.46 | 47.15 ± 21.26 |
| mnist / adv1 | activation | 0.9 | 47.14 ± 21.45 | 47.15 ± 21.26 |
| mnist / adv1 | activation | 0.99 | 46.85 ± 19.96 | 47.15 ± 21.26 |
| mnist / adv2 | constant | 0.9 | 38.01 ± 25.23 | 52.62 ± 19.26 |
| mnist / adv2 | constant | 0.99 | 3.42 ± 2.02 | 52.62 ± 19.26 |
| mnist / adv2 | linear | 0.9 | 51.68 ± 20.10 | 52.62 ± 19.26 |
| mnist / adv2 | linear | 0.99 | 48.00 ± 20.01 | 52.62 ± 19.26 |
| mnist / adv2 | activation | 0.9 | 51.32 ± 20.57 | 52.62 ± 19.26 |
| mnist / adv2 | activation | 0.99 | 48.97 ± 22.09 | 52.62 ± 19.26 |
| cifar10 / clean | constant | 0.9 | 70.07 ± 51.24 | 105.59 ± 39.01 |
| cifar10 / clean | constant | 0.99 | 1.13 ± 0.45 | 105.59 ± 39.01 |
| cifar10 / clean | linear | 0.9 | 102.17 ± 41.58 | 105.59 ± 39.01 |
| cifar10 / clean | linear | 0.99 | 78.52 ± 47.39 | 105.59 ± 39.01 |
| cifar10 / clean | activation | 0.9 | 101.73 ± 41.98 | 105.59 ± 39.01 |
| cifar10 / clean | activation | 0.99 | 76.32 ± 47.91 | 105.59 ± 39.01 |
| cifar10 / adv1 | constant | 0.9 | 76.68 ± 48.99 | 104.14 ± 39.76 |
| cifar10 / adv1 | constant | 0.99 | 7.61 ± 3.37 | 104.14 ± 39.76 |
| cifar10 / adv1 | linear | 0.9 | 104.65 ± 40.16 | 104.14 ± 39.76 |
| cifar10 / adv1 | linear | 0.99 | 91.70 ± 39.66 | 104.14 ± 39.76 |
| cifar10 / adv1 | activation | 0.9 | 104.66 ± 40.50 | 104.14 ± 39.76 |
| cifar10 / adv1 | activation | 0.99 | 80.01 ± 44.96 | 104.14 ± 39.76 |
| cifar10 / adv2 | constant | 0.9 | 79.17 ± 50.59 | 100.01 ± 42.77 |
| cifar10 / adv2 | constant | 0.99 | 5.04 ± 2.34 | 100.01 ± 42.77 |
| cifar10 / adv2 | linear | 0.9 | 99.11 ± 43.31 | 100.01 ± 42.77 |
| cifar10 / adv2 | linear | 0.99 | 82.03 ± 43.09 | 100.01 ± 42.77 |
| cifar10 / adv2 | activation | 0.9 | 98.37 ± 43.80 | 100.01 ± 42.77 |
| cifar10 / adv2 | activation | 0.99 | 80.95 ± 43.62 | 100.01 ± 42.77 |

Table 3: Results for the architecture E.

| Dataset / Training | Pruning | $\gamma$ | Avg time with pruning | Avg time w/o pruning |
|---|---|---|---|---|
| mnist / adv1 | constant | 0.9 | 81.98 $\pm$ 49.79 | 105.37 $\pm$ 38.76 |
| mnist / adv1 | constant | 0.99 | 3.31 $\pm$ 0.65 | 105.37 $\pm$ 38.76 |
| mnist / adv1 | linear | 0.9 | 106.47 $\pm$ 37.83 | 105.37 $\pm$ 38.76 |
| mnist / adv1 | linear | 0.99 | 104.06 $\pm$ 37.88 | 105.37 $\pm$ 38.76 |
| mnist / adv1 | activation | 0.9 | 104.43 $\pm$ 39.81 | 105.37 $\pm$ 38.76 |
| mnist / adv1 | activation | 0.99 | 91.64 $\pm$ 42.85 | 105.37 $\pm$ 38.76 |
| mnist / adv2 | constant | 0.9 | 89.78 $\pm$ 48.60 | 105.87 $\pm$ 38.62 |
| mnist / adv2 | constant | 0.99 | 3.20 $\pm$ 0.33 | 105.87 $\pm$ 38.62 |
| mnist / adv2 | linear | 0.9 | 106.25 $\pm$ 38.77 | 105.87 $\pm$ 38.62 |
| mnist / adv2 | linear | 0.99 | 103.36 $\pm$ 38.83 | 105.87 $\pm$ 38.62 |
| mnist / adv2 | activation | 0.9 | 106.77 $\pm$ 37.49 | 105.87 $\pm$ 38.62 |
| mnist / adv2 | activation | 0.99 | 102.95 $\pm$ 37.49 | 105.87 $\pm$ 38.62 |
| cifar10 / adv1 | constant | 0.9 | 128.13 $\pm$ 73.90 | 158.54 $\pm$ 58.54 |
| cifar10 / adv1 | constant | 0.99 | 7.26 $\pm$ 0.87 | 158.54 $\pm$ 58.54 |
| cifar10 / adv1 | linear | 0.9 | 158.95 $\pm$ 57.94 | 158.54 $\pm$ 58.54 |
| cifar10 / adv1 | linear | 0.99 | 159.39 $\pm$ 56.81 | 158.54 $\pm$ 58.54 |
| cifar10 / adv1 | activation | 0.9 | 162.66 $\pm$ 60.07 | 158.54 $\pm$ 58.54 |
| cifar10 / adv1 | activation | 0.99 | 158.40 $\pm$ 58.12 | 158.54 $\pm$ 58.54 |
| cifar10 / adv2 | constant | 0.9 | 135.34 $\pm$ 72.96 | 158.93 $\pm$ 57.85 |
| cifar10 / adv2 | constant | 0.99 | 3.73 $\pm$ 0.14 | 158.93 $\pm$ 57.85 |
| cifar10 / adv2 | linear | 0.9 | 159.26 $\pm$ 58.17 | 158.93 $\pm$ 57.85 |
| cifar10 / adv2 | linear | 0.99 | 158.93 $\pm$ 57.03 | 158.93 $\pm$ 57.85 |
| cifar10 / adv2 | activation | 0.9 | 159.35 $\pm$ 58.24 | 158.93 $\pm$ 57.85 |
| cifar10 / adv2 | activation | 0.99 | 158.48 $\pm$ 57.92 | 158.93 $\pm$ 57.85 |