# OpenReview forum: "Robustness Evaluation Using Local Substitute Networks"
_ICLR.cc/2024/Conference — ICLR 2024 Conference Withdrawn Submission_

### Official Review · Reviewer_kiZT · 2023-10-29

**Soundness:** 2 fair
**Presentation:** 1 poor
**Contribution:** 1 poor
**Rating:** 3
**Confidence:** 4

**Summary:**

An interval bound propagation style mechanism that considers the attempts to employ pruned surrogate models as a way to minimise the computational cost associated with robustness mechanisms.

**Strengths:**

The motivation of the authors is an important one - considering the computational implications of robustness mechanisms is an important part of moving these approaches towards utilisation in real world systems

**Weaknesses:**

The following notes have been provided to help the authors improve the quality of their manuscript. As much as a non-idiomatic command of the English language should not be a barrier to producing good science, it should be clear that these notes are not just focussed upon the language. Rather they also include points relating to the contributions, which I do not think are well established or contextualised. In my eye this paper in its current form is not suitable for publication.

I apologise to the authors for the volume of comments presented here, however I hope that they are taken in the helpful spirit that they have been provided with, and that they can be used to help improve your future scientific endeavours.

Contribution
- Establish robustness on smaller substitute networks saves computatioanl time. This, I do not believe, would be a surprise to anyone in the field - the question is not that this would make for faster calculations, but rather if this is accurate, when such an approach breaks down, and what the nature of the tradeoff is between size / architectural similarity / cost.
- "two networks with similar decision boundaries around the same anchor point must have similar robustness properties" - except this requires a) knowledge that the decision boundaries are similar and b) to be able to measure what a "similar" decision boundary or robustness property even means.
- If I'm reading this paper correctly, the pruning process is input dependent - that each individual sample requires a different pass through the pruning process. If the argument here is one of efficiency, then that need to repeatedly prune would seem to be wildly inappropriate.


Abstract:
- "To assess the robustness," - if it is "the robustness" then it's the robustness of something. Framing like this without a subject doesn't make sense.
- "For the smaller local" - there is not a singular smaller network that exists, so it is "a smaller local" rather than "the".
- Figure 1 "The sample, we are interested in, is marked by" - no need for the commas here.
- Figure 1 " While the global behaviour changes a lot...remains similar" - what is meant by a lot here, and similar? What is the anchor point?
- Figure 1 "does not change significantly" - such a statement is highly contextual, but without any metrics or contextualisation this is meaningless.


Introduction:
- "Irreplaceable" is a rather strong statement without evidence. You're assuming here that no other alternatives can match the performance of NN's (which I think is an unfounded statement when you consider the scope of potential applications), and in this sentence you're also assuming that practitioners know of the flaws in these networks, which is not always true.
- "Containuously gains more attention" - more introduces this as a relative construct, but you haven't established what is receiving more attention than. Plus, needs citations.
- "small perturbations of the correctly classified samples that lead to a false prediction." Few points here, samples is plural, prediction is singular, so there's a confused context here. Labelling this as "a correctly classified sample" would make it clearer that a) singular and b) that the sample being correctly classified isn't an intrinsic property of the point, but rather the subject of manipulation. More broadly there's also a structural issue regarding the subject - in the sentence as it stands is it the small perturbations that produce the false predictions (no), or is it the correctly classified samples (also no). The subject of your sentence being the perturbed versions of input samples is not clearly established, so then discussing false predictions is meaningless.
- "where the data is collected from sensors" - how does this contextually link to the rest of the sentence? Sensor collected data is not intrinsicly linked to the datasets you've previously established, nor to the risks of adversarial examples.
- "it is not acceptable that the same image, for example, a road sign, is classified differently depending on the signal noise." None of your subject to this point has been the signal noise. Your subject to this point has been adversarial examples which are constructed with perturbations (which is crafted), which differs completely from the uncrafted signal noise context. Also "not acceptable" implies a value judgement - not acceptable to whom? If you are to maintain the focus on signal corrupted images, I'd suggest here focussing upon images that are semantically identical, but whose difference is driven by the addition of sensor noise.
- "While this problem is widely known" - the subject of this is not clear? Is this signal noise? Adversarial examples? A classifier predicting different outputs for the same input? A statement like "this problem" assumes that the reader is completely aware of what the context is, but because the previous sentence jumps between subjects, the actual nature of the problem is not clear here.
- It's odd that an introduction that leans so heavily on the concept of robustness only cites Katz. There's a large amount of robustness literature that could be touched on here.
- "Therefore, constructing adversarial attacks and measuring the perturbation's magnitude, which is required to change the prediction, is still one of the most popular ways to estimate the network's robustness". Okay, so I'm going to break down my response here to multiple parts
a) Too many unneccessary commas. Something like "Therefore the size of the perturbations required to change the prediction...." reads so much cleaner.
b) Because you moved your context of adversarial examples from perturbations to sensor corruption in the prior paragraph, the nature of what the perturbation is isn't clearly established.
c) "the networks" - should be "a networks", because there is not one specific network being considered here.
d) The statement regarding attacking to establish robustness isn't true! It is well established that attacking a model is a particularly poor way of measuring the smallest possible adversarial attack that can exist, due to the long history of advances in attacks. This is why methods like randomised smoothing and interval bound propagation exist.
- Why is your enumerated list of contributions presented like this? At the very least capitalise the first letter of each block, and use a colon to start. At the moment this is essentially the worlds longest sentence.
- The second to last paragraph is a restatement of the Figure caption. Why are both needed in similar levels of detail?

Notation:
- "we develop our analysis for the classification networks that Li et al (2023) use for their comprhensive overview" - so....what is it that you're doing that's  different to Li? How do classification networks come into things here? And that's beyond the strange structure.
- "since most of the verification methods support it" -> "since most verification methods support it". This is a point that would be a lot easier to make if you had cited any verification methods, or if your introduction hadn't broadly focused upon attacks as a verification method.
- The nature of what an anchor point isn't hasn't been established. This is not a standard bit of notation.
- What is different to the content presented here to interval bound propagation?

Local substitute:
"We must reduce its size" - must we? Why must we? There are other alternatives here.
- "Practitioners pursue the same goal when the network has to be deployed in a setting which strictly constrained available memory of when the forward pass during inference must be particularly fast." This is a word salad. What practitioners? What goal? This sentence is about twice as long as it needs to be. If you're concerned about a memory constrained environment, establish that as part of your motivating context, this is way too late in the document for it to come up now.
- "by removing its neurons and weights" - well, if you remove the neurons and the weights then there's no network. You remove a subset of the neurons and weights.
- "drop in performance onthe main task" by what metric? What is the main task?
- "and even improving generalization to new data" - this is a whole separate context, don't throw it into the sentence at this point.
- "the primary goal - faster verification- we" - inconsistent spacing around the dash's. In tex use "goal---faster verification---we" to use the proper em-dash.
- "the worse the possible approximations" what approximations? Context is not clear.
- " for the nonlinear activation is the simplest option to reduce the complexity of the verification problem" - what does this even mean? "for the nonlinear activation is the simplest" is confusing enough, but the context of what you're actually talking about here is so opaque.
- "we do not only ensure that the network's output at the base point x0 remains the same" - that this is between the base and pruned classifiers isn't clear. And doing this doesn't establish that the decision boundary remains constant, it just means that, if successful, the decision boundary exists at some point between x0 and x0'. But the size of an adversarial attack can be quite large, so this is a very loose bound on the behaviour of your two networks.
- "Bounds propagation techniques, as defined in Section 2, are widely used in [the] verification literature" THEN WHERE ARE THE CITATIONS? To this point you've only cited Katz and Le.
- "like sigmoid" -> "like the sigmoid". But other bound propgataion techniques have (if memory serves) shown it is possible to bound sigmoid style functions.
- Algorithm 1 and associated content "by pruning the least uncertain neurons" - where is this within Algorithm 1?

Experiments:
- So far you have two citations. At least cite the authors of MNIST and CIFAR-10.
- "datasets.We call" - appears to not have a space after the full stop.
- "We call them according to authors' notation C-E, where C...." - this is completely nonsensical and un-parsable.
- "Algorithms A, B, and D" what are these?
- "All based on the unified toolbox by Li et. al. (2023) that will be publicly available" - I'm curious about if this is a potential break of double-blind. If Li et. al's code was not published, then I would be 100% confident that the authors overlap. After chasing it down, it appears that Li's code is published, so I'm going to be generous and not read this sentence as it is written, which implies that both pieces of work share the same nonpublic code, and that code is pending release. Otherwise this would be a break of double blind.
- One of your arguments earlier on in the paper was that pruning is needed to manage GPU vram and computational cost more broadly. Yet this isn't addressed in your results or discussion - there is a column of results covering the computational time with and without pruning - but in its current form it's not clear the level of effect that can be attributed to the pruning process.

Related work:
- You define mixed integer linear programming as MILP here. Before this point you'd used MILP as a term 6 times without it being defined.

Future work:
- The first paragraph is a restatement of the comment about Katz in the introduction. Why are both needed?

At this stage I think I've provided enough feedback, so the density of my comments has reduced, but I would close by stating that currently the results as presented are meaningless - they are not contextualised in a way that would help one understand what is important, what the contributions that can be drawn from them, or if there are limitations or drawbacks associated with the pruning process.

Oh, and the bibliography is a mess of mixed cases.

**Questions:**

I apologise, but I do not believe that there is anything in my comments above that would lead me to changing my review at this time. I hope that the authors take the comments attached above and use them as a guide to improving their future works.

---

### Official Review · Reviewer_eJVB · 2023-10-30

**Soundness:** 3 good
**Presentation:** 2 fair
**Contribution:** 2 fair
**Rating:** 3
**Confidence:** 4

**Summary:**

The paper presents a method for estimating the adversarial robustness of neural
networks using analyses on smaller, substitute networks. A substitute network
is obtained by replacing the ReLU functions by either a constant or a linear
function, which are easier to handle by a neural network verifier.

**Strengths:**

The paper makes the empirical observation that the adversarial distance on the
smaller model can at times be (relatively) close to the distance from the
original model.

**Weaknesses:**

The main weakness is that no formal connection is established between the
adversarial distance obtained by the simplified model and the corresponding
distance on the original model. As a result the output of the proposed method
cannot be generally trusted.

The proposed method is not compared with the cited work on linear
approximations. The latter can be used very effectively whilst formally
bounding the adversarial distance.

The number and sizes of the networks considered in the empirical evaluation are
not reported.

**Questions:**

See comments above.

---

### Official Review · Reviewer_7VfR · 2023-11-01

**Soundness:** 3 good
**Presentation:** 3 good
**Contribution:** 3 good
**Rating:** 8
**Confidence:** 3

**Summary:**

The paper introduces an innovative approach to accelerate the formal verification of deep neural networks. The authors present a pruning technique that selectively substitutes non-linear activation functions with linear alternatives. This substitution is particularly effective as non-linear activation functions are a primary contributor to the computational cost associated with verification tasks. The pruning process is carefully designed to maintain the robustness and local behavior of the original network as closely as possible. Consequently, the verification certificate of the pruned network effectively represents that of the original network.

**Strengths:**

1. The paper exhibits exceptional writing quality. It offers a clear and comprehensible narrative, effectively conveying complex technical concepts in an articulate manner. It also ensures that all essential background information is provided, making it accessible even to readers who may not possess an in-depth familiarity with the subject matter.
2. I personally find the concept of network pruning for the purpose of accelerating verification quite appealing. While pruning techniques have seen widespread use in various areas of machine learning and computer vision, the author's application appears to be genuinely innovative, as far as my knowledge extends.
3. The focus on preserving the network's robustness properties by considering its local behavior during the pruning process is a clever and intuitive approach. It demonstrates a thoughtful consideration of the network's behavior, which adds a valuable dimension to the proposed methodology.

**Weaknesses:**

1. Runtime comparison is not thorough. Particularly, in case of the proposed method, the pruning time is not included in the total time required for vertification.
2. Effectiveness of the method has been only demonstrated using a single verification method, ie, using MILP solvers. The literature on formal verification has a lot of different methods, and MILP-based is not even the most popular one. Even though authors comment that the effectiveness of their method is agnostic to the verification approach, I believe adding empirical results to support this claim are necessary.

**Questions:**

1. How much does the pruning time contribute to the total runtime? Specifically, how does the pruning time compare to the time required for solving the MILP task?
2. Does the proposed method work for other verification methods? How about probabilistic methods like randomized smoothing?

---

### Official Review · Reviewer_iXnZ · 2023-11-01

**Soundness:** 2 fair
**Presentation:** 2 fair
**Contribution:** 2 fair
**Rating:** 3
**Confidence:** 4

**Summary:**

This work presents a neural network robustness evaluation method based on network pruning. This method speeds up the evaluation process by pruning the origin network to form a smaller local substitute network, thereby saving computation.

**Strengths:**

This paper is well-organized, offering theoretical explanations and a comprehensive literature review.

**Weaknesses:**

1. Insufficient experimental evaluation

(1) In Table 1 and Appendix C, for different network structures, the authors present the speedup performance of the proposed method on the MILP-based verification approach. However, verification methods with better certification performance and higher efficiency exist. For example, LBP-based verification methods [1] and [2]. While the authors offer a theoretical demonstration of robustness guarantees in Appendix A, it is strongly recommended that they experimentally evaluate the proposed method on these SOTA methods to confirm its adaptability.

(2) Existing methods also utilized pruning-based techniques to enhance the certified robustness evaluation, such as [3]. To confirm the effectiveness and superiority of the proposed method, it is strongly recommended that the authors carry out comparative evaluations with these methods.

(3) According to the Experiment Setup section, the authors assessed the speed-up performance of the proposed method on the benchmark certificated robustness evaluation method using three convolutional neural networks used in [4] (referred to as network C-E in this paper). However, the authors of [4] also evaluated a larger convolutional network with 17247946 parameters on the CIFAR-10 dataset. This network’s evaluation is absent in the experiment section.

2. Limited applicability.

The pruning methods proposed in this paper aim to improve the efficiency of the robustness evaluation. However, the evaluation results in the experiment section only show the mean absolute difference between the predicted and original decision boundaries, which doesn’t necessarily ensure reasonable precision for evaluation bounds. Given that evaluation accuracy is one of the most essential metrics for robustness evaluation, emphasizing speed at the expense of this accuracy might cause the model to fail unexpectedly. Therefore, this reviewer is concerned about the broad applicability of this method.


[1] Wang S, Zhang H, Xu K, et al. Beta-crown: Efficient bound propagation with per-neuron split constraints for neural network robustness verification[J]. Advances in Neural Information Processing Systems, 2021, 34: 29909-29921.

[2] Zhang H, Wang S, Xu K, et al. General cutting planes for bound-propagation-based neural network verification[J]. Advances in Neural Information Processing Systems, 2022, 35: 1656-1670.

[3] Sehwag V, Wang S, Mittal P, et al. Hydra: Pruning adversarially robust neural networks[J]. Advances in Neural Information Processing Systems, 2020, 33: 19655-19666.

[4] Li L, Xie T, Li B. Sok: Certified robustness for deep neural networks[C]//2023 IEEE Symposium on Security and Privacy (SP). IEEE, 2023: 1289-1310.

**Questions:**

1. Can the authors experimentally evaluate the effectiveness of the proposed method on the SOTA robustness evaluation methods, such as [1] and [2], to confirm its adaptability?

2. Why did the authors not include a robustness evaluation on the larger convolutional network, as mentioned in [4], in their experiment section?

3. Can the authors comparatively evaluate the proposed method with other pruning-based evaluation methods, such as [3], to confirm the effectiveness and superiority of this method?

4. How can the authors ensure reasonable precision for evaluation bounds when emphasizing speed in the proposed method?

---

### Official Review · Reviewer_wHPv · 2023-11-02

**Soundness:** 2 fair
**Presentation:** 3 good
**Contribution:** 2 fair
**Rating:** 5
**Confidence:** 2

**Summary:**

This paper empirically shows that compared to directly evaluating the original networks, computing the metrics for the smaller local substitute networks yields reasonable estimates of the robustness for a lower cost. To construct the substitute network, the authors develop several pruning techniques that preserve the local properties of the initial network around a given anchor point. Experiments are done on multiple datasets, showing that this approach can save computation and is beneficial for larger models.

**Strengths:**

The strengths of this paper include:
- The writing is clear with intuitive explanation on the adversarial mechanism and verification processes.
- The motivation is reasonable that reducing computational cost is critical for applying model verification in scalable settings.
- The save on empirical computational time seems significant (although the results in, e.g., Table 1, are somewhat hard to parse).

**Weaknesses:**

The weaknesses of this paper include:
- Is there any formal guarantee that model verification on local substitute networks is a good estimation (with theoretical bounds) of it on original network?
- The authors claim that their method is especially beneficial for larger models, however, the models evaluated in experiments (described in the Setup paragraph in Section 4) are relatively small. Can the proposed method scale to more practical architectures such as WRN-34-10?

**Questions:**

I need to admit that I'm familiar with adversarial attacks, but I'm not an expert on model verification. So as far as I understand, my main concerns are: 1. is there formal guarantee on the connection between model verification on local substitute networks and it on the original network? 2. Empirical evaluations should be done on practical model architectures.

---

### Official Review · Reviewer_D5Te · 2023-11-02

**Soundness:** 2 fair
**Presentation:** 2 fair
**Contribution:** 2 fair
**Rating:** 3
**Confidence:** 2

**Summary:**

This paper addresses the problem of the runtime complexity of computing robustness certifications on an approximate network which replaces nonlinear activations with linear activations.

**Strengths:**

- experimental results demonstrate a large drop in runtime for computing verifications with proposed approach
- experimental results and some theoretical results in appendix to demonstrate that approach gives close certified radii
- good scope in experiments: multiple datasets and architectures

**Weaknesses:**

- clarity can be improved a bit.  The authors use different mathcal letters to represent algorithms, but it may be more clear to just specify the algorithm name rather than using the mathcal letter.  It would also help if the authors specified the architectures used within the appendix.
- experiments in appendix on other architectures do not show the gap between the predicted verification bound and the actual bound
- in a lot of cases for CIFAR-10 the difference in predicted bound seems quite large relative to the perturbation size used

**Questions:**

- I confused about the set up, since the activations are replaced depending on a single test example, are pruned models generated for each test example?  Or is there only 1 pruned model?